# The Saint-Leonard Urban Glaciotectonic Cave Harbors Rich and Diverse Planktonic and Sedimentary Microbial Communities

**DOI:** 10.3390/microorganisms12091791

**Published:** 2024-08-29

**Authors:** Jocelyn Lauzon, Daniel Caron, Cassandre Sara Lazar

**Affiliations:** 1Biological Sciences Department, University of Quebec in Montreal (UQAM), Montreal, QC H3C 3P8, Canada; 2Spéléo Québec, Montreal, QC H1S 2A7, Canada

**Keywords:** microbial ecology, bacteria, archaea, eukaryotes, ultra-small microorganisms, subsurface, caves, glaciotectonics, urban ecology

## Abstract

The terrestrial subsurface harbors unique microbial communities that play important biogeochemical roles and allow for studying a yet unknown fraction of the Earth’s biodiversity. The Saint-Leonard cave in Montreal City (Canada) is of glaciotectonic origin. Its speleogenesis traces back to the withdrawal of the Laurentide Ice Sheet 13,000 years ago, during which the moving glacier dislocated the sedimentary rock layers. Our study is the first to investigate the microbial communities of the Saint-Leonard cave. By using amplicon sequencing, we analyzed the taxonomic diversity and composition of bacterial, archaeal and eukaryote communities living in the groundwater (0.1 µm- and 0.2 µm-filtered water), in the sediments and in surface soils. We identified a microbial biodiversity typical of cave ecosystems. Communities were mainly shaped by habitat type and harbored taxa associated with a wide variety of lifestyles and metabolic capacities. Although we found evidence of a geochemical connection between the above soils and the cave’s galleries, our results suggest that the community assembly dynamics are driven by habitat selection rather than dispersal. Furthermore, we found that the cave’s groundwater, in addition to being generally richer in microbial taxa than sediments, contained a considerable diversity of ultra-small bacteria and archaea.

## 1. Introduction

The Earth’s terrestrial subsurface harbors an important microbial diversity whose community structures, metabolic pathways and ecological functions reflect the peculiarities of their environment [1,2,3]. Bacteria, archaea, microeukaryotes and viruses striving in the deep continental biosphere could account for up to one-fifth of the planet’s microbial biomass, weighing around 10^16^ to 10^17^ g C [4]. The subsurface comprises a still largely unexplored array of habitats supporting microbes—e.g., aquifers [5], bedrock [6], caves [7], abandoned mines [8,9] and petroleum reservoirs [10]—in which harsh conditions often prevail [11], such as total darkness, low-nutrient input levels, variable water availability and humidity, high or low temperatures and/or anoxic conditions. Despite the challenges associated with sampling belowground like limited accessibility and potential contamination by heavy equipment [2,12], the study of subterranean microbiology has emerged by the turn of the century as a promising field [13]. The continental subsurface provides opportunities for extreme environment adaptation exploration [14], including their repercussion on biotic interactions. One example of these adaptations is the ultra-small prokaryotes, which generally strive in aquifers since smaller sizes are linked to a response strategy to harsh environmental conditions [15,16]. Ultra-small bacteria and archaea belonging to CPR and DPANN superphyla have been detected in groundwater [17]. These ultra-small microorganisms have a volume of ≤0.1 µm^3^ (diameter from <0.05 to 0.40 µm), a streamlined genome (0.58 à 3.2 Mbp) and show a loss of facultative and essential metabolic pathways. Consequently, they often form symbiotic associations with other prokaryotes [15,18], and an interkingdom symbiosis has even been described [19]. The study of subsurface microorganisms has also led to the discovery of metabolisms capable of bioremediation [20,21,22]. Furthermore, research on these peculiar microbial communities supports a better understanding of global biogeochemical cycles [2,23], early forms of life on earth [24] and the rise of multicellularity [25]. The subsurface also provides opportunities to make hypotheses about characteristics of life elsewhere in the universe [26,27].

A component of subsurface ecosystems is caves, which are especially interesting habitats to investigate. As natural cavities in rocky environments, they constitute unique passages, or “windows”, for the scientists to access the underground world and its inhabitants [28]. Barton and Jurado [29] hypothesized that due to the poor nutrient state of their environment and the often-limited metabolic capacities of species, underground microbial community dynamics might be driven by mutualistic associations rather than the exclusion of competitors. As demonstrated by Anantharaman et al. [30] in an aquifer ecosystem, syntrophic interactions are essential to subterranean microbial community functioning since few microorganisms can perform all redox reactions of a given pathway. Nonetheless, some cave microorganisms produce antimicrobials, and their genomes host antimicrobial resistance genes [31]. These bioactive secondary metabolites mediate microbe interactions, acting as weapons in interference competition but also as signaling molecules and as a food source [32,33,34]. *Actinobacteriota*, which abound in cave environments, comprise taxa that naturally produce antimicrobial metabolites, some of which are used in medicine [35].

In addition, caves often provide diverse interrelated ecological compartments—notably water and sediments—calling for particular taxonomic and functional associations. Most caves are oligotrophic and static environments [7,36], yet molecular phylogenetic techniques have shown that these caves can be home to unique taxa and assemblages of prokaryotic and eukaryotic microorganisms [37,38,39]. The Bacteria domain is the most abundant domain found in cave ecosystems, and the lineages found in oligotrophic caves span the entire domain [29], with *Actinobacteriota* and *Proteobacteria* phyla generally dominating [40,41,42]. Archaea are mainly represented by members of *Crenarchaeota* [38,43,44]. As for the eukaryotes, fungi—especially the *Ascomycota* phylum [45]—as well as organisms from the *Alveolata* clade and the TSAR supergroup [46], generally contribute to the microbial community structure of the different ecological compartments of caves.

If the extreme abiotic conditions shape the biotic communities of caves, the microbes can, in turn, impact their environment through various destructive and constructive processes [13,47]; for instance, the dissolution of carbonate rock by secretion of sulfuric acid [48] and the formation of speleothems—secondary mineral deposits—such as moonmilk [49], pool fingers [50] and coralloids [51]. Owing to the myriads of metabolisms sustained by cave microorganisms, microbial communities are critical participants in global geochemical cycles of key elements, namely carbon, nitrogen, sulfur [41,52], iron [53,54], manganese [44,55] and phosphorus [56]. Caves are predominantly formed in a karst, a geologic environment or landscape of extensive groundwater flow systems where speleogenesis occurs through the dissolution of soluble rock, mainly carbonates such as limestone [57,58]. But the formation of caves can also naturally result from several other geological and chemical processes [59], among which is the mechanical action of glacier movement on sedimentary rock [60]. Such glaciotectonic caves have so far only been detected in the province of Quebec, Canada [61]. The Saint-Leonard cave is located on the island of Montreal (Quebec, Canada). Its glaciotectonic origin traces back to around 13,000 years ago, during the withdrawal of the Laurentide Ice Sheet. This non-karstic cave was mechanically formed by the dislocation of the 460 M-year-old sedimentary rock composing the Rosemont Member of the Montreal Formation (Trenton Group, Ordovician) in the St. Lawrence Lowlands physiographic region [61]. The sub-horizontal clayey limestones interstratified with thin calcareous shales were subjected to pressure, thrust and friction from the moving continental glacier, thus resulting in the opening of preexisting fractures and the interbed sliding of strata along shales, with the upper layers displacing further than the lower ones [60,62].

The unusual glaciotectonic speleogenesis of the Saint-Leonard cave, combined with its urban setting, makes it a unique environment of the planet’s subsurface that has yet to be biologically explored. By using amplicon sequencing of the 16S/18S rRNA genes for the Bacteria, Archaea and Eukaryote domains, this metataxonomic study aims to analyze the microbial communities’ structure present in the water column and sediments of the cave, as well as their links to surface microbial communities potentially seeping inside the cave. As a first exploratory inquiry of bacterial, archaeal and eukaryotic life of Saint-Leonard’s underground galleries, our main objective was to characterize and compare the taxonomic diversity and composition of aquatic and sedimentary communities, as well as those from the surface soils, for each of the three domains. Special attention was given to ultra-small prokaryotic taxa living in the cave’s water since they have been shown to compose a substantial part of groundwater communities.

## 2. Materials and Methods

### 2.1. Study Site Description

The entrance to the Saint-Leonard cave is located in the municipal Pie-XII Park in Montreal, Quebec, Canada (45.588234 N 73.607872 W) (Figure 1a). The cave was discovered in 1812 but only 35 m of corridor were then known to exist. Excavation work in 2017 revealed 370 m of new galleries [61]. The cave is partially open to the public, who can book guided visits to explore the historic portion only. Overall, the galleries of the cave extend over 400 m long with a ceiling 3 to 5 m deep belowground from which tree roots hang in some places. The corridors’ width is between 0.5 to 4 m and their height can reach up to 7 m at some points. Two galleries compose the cave: Radiesthesia Gallery has a partly dry floor consisting mainly of rocky debris, while Echo Gallery is entirely aquatic and needs to be navigated by kayak [62]. The depth of the water table inside the Echo Gallery’s corridors fluctuates with local and seasonal hydrological events like snow melting and rainfall and can be more than 4 m high [60]. Sediments have deposited on the bottom floor and also on numerous edges of the walls due to the oscillation of the water level [60]. Water residence time is short [63]. Air and water temperature varies seasonally from 5 to 12 °C [62].

### 2.2. Sampling and Water Filtration

For amplicon sequencing analyses, sampling of water (W) and sediments (S) inside the cave was performed in October 2021 in the two galleries (Radiesthesia and Echo) closed to the public (Figure 1b). Seven water samples (two from the Radiesthesia and five from the Echo) were collected in sterile polypropylene bottles (Nalgene, Rochester, NY, USA) and four sediment samples (three from the Radiesthesia and one from the Echo) were collected in 50 mL sterile Falcon tubes. Samples were kept at 4 °C during transportation to the laboratory. Upon arrival, water was immediately filtered, and sediments were frozen at −80 °C. Each 500 mL water sample was separated into two size fractions. First, the water was filtered through a 0.2 µm (W2) polyethersulfone filter (Sartorius, Midisart, Germany) with a vacuum pump. Filtrate was collected in a sterile Erlenmeyer, then filtered a second time through a 0.1 µm filter (W1). Filters were kept frozen at −80 °C until DNA extraction.

Eight surface soil samples (SS) were collected in the parks surrounding the entrance to the cave, all potential water seepage sources for the groundwater in the cave (Figure 1c). One surface water puddle (SW) close to the soil sample SS8 was also collected in a 50 mL sterile Falcon tube. The soil samples were stored at −80 °C upon arrival at the lab, and the water sample was filtered as described above.

### 2.3. Water Geochemical and Physicochemical Analyses

Subsamples of the water collected inside the cave were used for physicochemical analysis. For dissolved organic and inorganic carbon (DOC/DIC), water was filtered through a 0.45 µm polyethersulfone filter (Sarstedt, Numbrecht, Germany) and stored in gas-free glass bottles, which were kept at 4 °C until analysis. Water was filtered through a 0.22 µm filter for ammonia/ammoniac (NH_X_) and through a 0.45 µm filter for nitrate (NO_3_^−^) and nitrite (NO_2_^−^), then collected in plastic scintillation bottles, which were frozen at −20 °C. All further analyses were conducted at the GRIL (Interuniversity Research Group in Limnology)—UQAM (Université du Québec à Montréal) analytical laboratory. DOC and DIC concentrations (mg/L) were measured with an Aurora 1030 W TOC Analyzer (OI Analytical, College Station, TX, USA) using a persulfate oxidation method. An OI Analytical Flow Solution 3100 continuous flow analyzer was used to measure inorganic nitrogen concentrations (mg/L). Ammonia/ammonium was quantified using a chloramine reaction with salicylate to form indophenol blue dye (EPA Method 350.1). Nitrate and nitrite were quantified using an alkaline persulfate digestion method, coupled with a cadmium reactor, following a standard protocol [64]. pH and temperature were measured on-site with a YSI multiparameter probe (model 10102030, Yellow Springs, OH, USA).

### 2.4. Sediment and Surface Soil Characteristics

Subsamples of the surface soil and cave sediment samples were dried for 72 h at 22 °C under a laminar flow hood, then finely powdered with a mortar and pestle. All physicochemical analyses were conducted at the GEOTOP laboratory (UQAM). Total carbon (Ctot), organic carbon (Corg), inorganic carbon (Cinorg) and total nitrogen contents (Ntot) were measured with a Carlo Erba NC2500 elemental analyzer (Thermo Fisher Scientific, Waltham, MA, USA). Prior to organic carbon measurements, samples were fumigated with hydrochloric acid for 24 h to eliminate inorganic carbon. Carbon-13 (^13^C) and nitrogen-15 (^15^N) isotope contents were measured by stable isotope ratio mass spectrometry with a Micromass Isoprime 100 spectrometer, coupled to a Vario MicroCube elemental analyzer (Elementar, Lyon, France) in continuous flow mode. To measure pH, sediment powder was suspended in ultra-pure double-deionized water (Milli-Q) in a 1:4 ratio and mixed continuously for 30 min. pH values were obtained with a combined glass electrode (accuTupH and Accumet XL600; Thermo Fisher Scientific).

### 2.5. DNA Extraction

DNA was extracted from water filters (0.2 and 0.1 µm) with the DNeasy PowerWater Kit (Qiagen, Hilden, Germany) following the manufacturer’s protocol. DNA was eluted in 100 µL 10 mM Tris HCl (pH 8.5) and stored at −80 °C. One negative control was prepared for each filter size by filtering 500 mL of autoclaved ultra-pure water and using it with the kit in the same conditions as the cave water samples. For surface soil and cave sediment samples, DNA was extracted with the DNeasy PowerSoil Kit (Qiagen) following the manufacturer’s protocol, eluted in 100 µL 10 mM Tris HCl (pH 8.5) and then stored at −80 °C. One negative control was prepared using sterile ultra-pure water with the kit.

### 2.6. PCR, Library Preparation and Sequencing

Polymerase chain reaction (PCR), library preparation and sequencing of 16S and 18S rRNA genes were performed at The Center of Excellence in Research on Orphan Diseases—Fondation Courtois (CERMO-FC, UQAM). PCR amplification was carried out using the Phusion Hot Start II DNA Polymerase (2 U/µL) (Thermo Fisher). For bacteria, the V3-V4 hypervariable region of the 16S rRNA gene was targeted using the B341F (5′–CCTACGGGAGGCAGCAG–3′) [65] and B785R (5′–GACTACCGGGGTATCTAATCC–3′) [66] primer pair. For archaea, the V3-V4-V5 region of the 16S rRNA gene was targeted using the A340F (5′–CCCTACGGGCYCCASCAG–3′) [67] and A915R (5′-GTGCTCCCCCGCCAATTCCT–3′) [68] primer pair. For eukaryotes, primer pairs E960F (5′–GGCTTAATTTGACTCAACRCG–3′) [69] and NSR1438R (5′–GGGCATCACAGACCTGTTAT–3′) [70] were used to target the V5-V7 region of the 18S rRNA gene. PCR was carried out following these conditions: denaturation at 98 °C for 30 s; annealing for 30 s at 57 °C for Bacteria, 67 °C for Archaea and 55 °C for Eukaryote domains; and extension occurred at 72 °C for 1 min. Final extension occurred at 72 °C for 10 min, after 35 amplification cycles for Bacteria and Archaea and 33 cycles for the Eukaryote domain. PCR products were normalized and purified, and libraries were submitted to quality control. Sequencing was performed on an Illumina MiSeq 2300 using the MiSeq Reagent Kit v3 (600 cycles; Illumina, San Diego, CA, USA). For each domain, a PCR-negative control was sequenced. Raw sequences were deposited on the National Center for Biotechnology Information platform (NCBI) under the BioProject ID PRJNA1139756.

### 2.7. Sequence Analysis

Amplicon sequence variants (ASV) were generated from raw sequences using DADA2 (v.1.24) [71] in R (v.4.2.2) [72]. Primers were removed and forward and reverse reads were truncated at positions 275 and 225 (bacteria) and at positions 260 and 230 (eukaryotes). For archaeal sequences, because of the low quality of the reverse reads, the forward and reverse reads could not overlap, and only the forward reads were kept and truncated at position 275. Sequences were then consolidated and denoised, and chimeras were removed to obtain an ASV table for each domain. We discarded the cave water sample #7 filtered at 0.1 µm due to a pre-PCR sample processing error resulting in an ASV composition extremely similar to 0.2 µm water samples for both prokaryote domains.

Taxonomic annotations of ASV were made using the SILVA SSU database (v.138.1) for the bacteria [73]. For the archaea, we used a personal database to further classify ASV from the *Bathyarchaeota* phylum (based on Zhou et al. [74]) and *Woesearchaeota* phylum (based on Liu et al. [75]). For the eukaryotes, we used the PR2 database (v.4.14.0) [76]. To decontaminate the ASV communities of each type of sample (water filtered at 0.2 and 0.1 µm, surface soils and sediments), we used the decontam package [77] with the kit blank control samples and the negative PCR controls, resulting in the removal of 174 contaminant ASVs for the Bacteria (3.2% of total bacterial ASVs), 11 contaminant ASVs for the Archaea (0.4% of total archaeal ASVs), and 39 contaminant ASVs for the Eukaryote (2.0% of total eukaryotes ASVs) domains. Finally, ASV tables were normalized using the median depth sequencing method [78].

### 2.8. Statistical Analysis

All statistical analyses were carried out using R (v.4.2.2) [72] unless otherwise stated. All statistical analyses described below were conducted for each of the three domains separately, and the statistical significance level was set to 0.05. We did not analyze amplicons for the 0.1 µm water samples for the Eukaryote domain. Shannon diversity indices, as well as richness (Chao1) and evenness diversities, were calculated to compare alpha diversity between the different habitats (water filtered at 0.1 µm, water filtered at 0.2 µm, sediments, and surface soils). Kruskal–Wallis and Dunn tests were performed using the dunnTest function of the FSA package [79].

To analyze beta diversity, we first visualized community compositional variation between samples by performing a Principal Coordinate Analysis (PCoA) on a Bray–Curtis dissimilarity matrix computed with PAST (v.4) software [80]. To test if community composition differed between habitat types, a PERMANOVA [81] was performed with the adonis2 function of the vegan package (nperm = 999) [82]. We used analysis of molecular variance (AMOVA) to further distinguish which sample group differences significantly explained the PERMANOVA results. We used homogeneity of molecular variance (HOMOVA) to test compositional homogeneity between two sample groups. Both these tests were run in mothur (v.1.47) [83] using the AMOVA and HOMOVA functions. Correlation between the Bray–Curtis dissimilarity matrices and environmental matrices was tested with a distance-based redundancy analysis (db-RDA), on a Hellinger-transformed ASV matrix using the capscale function of the vegan package in R. The significance of explanatory variables was assessed with the anova function with 200 permutations. The contributions of each significant variable were determined with the varpart function in vegan.

We constructed bar plots showing the relative abundance of phyla and genera in each sample and ran a Linear Discriminant Analysis Effect Size (LEfSe) to reveal which genera were the most likely to explain the compositional difference between habitat types, using the lefse function in mothur. To estimate the proportion of surface soil and water microbial communities (sources) contributing to the formation of the sediment and water microbial communities in the cave (sinks), we used fast expectation–maximization microbial source tracking (FEAST) [84].

## 3. Results

### 3.1. Water, Sediment and Soil Characteristics

Groundwater samples had homogeneous environmental conditions regarding DIC, DOC and inorganic nitrogen concentrations (ammonia, ammonium, nitrite and nitrate) (Table 1). We measured an average DOC concentration of 1.64 mg/L and an average DIC concentration of 46 mg/L. The water pH was close to neutral, with an average of 7.54.

Cave sediment samples showed variations in inorganic and organic carbon, which were both low, as well as total nitrogen content (Table 2). pH was slightly basic with an average of 8.1. δ^13^C values ranged from −26.3 to −24.7‰, and δ^15^N values were only obtained for samples S2 (7‰) and S4 (3.8‰). Apart from sample SS1, which was acidic, the surface soils had a pH varying from 7.1 to 7.8 and were characterized by a high content in organic carbon constituting most of the carbon pool. Total nitrogen concentrations were on average 6 times higher than in the cave sediments. δ^13^C values ranged from −28.1 to −24.4‰, and δ^15^N values ranged from 0.5 to 4‰.

### 3.2. Alpha-Diversity of the Microbial Communities

Bacterial Shannon indices did not significantly differ between the four different habitats (cave water 0.2 µm, water 0.1 µm, cave sediments and surface soils) (Appendix A). ASV richness was significantly different between cave sediments and water 0.2 µm (S > W2) and between water 0.2 and 0.1 µm (W1 > W2). Evenness was significantly different between cave sediments and surface soils (SS > S), surface soils and water 0.1 µm (SS > W1) and between water 0.2 and 0.1 µm (W2 > W1).

Archaeal Shannon indices were significantly different between surface soils and both cave water samples (W1 > SS, W2 > SS), as well as between the sediments and both water samples (W1 > S, W2 > S) (Appendix A). The same was observed for ASV richness except that sediment richness was not significantly different from water 0.1 µm (W1 > SS, W2 > SS, W2 > S). Evenness was significantly different between surface soils and water 0.1 µm (W1 > SS), sediments and water 0.1 µm (W1 > S) and between water 0.1 and 0.2 µm (W1 > W2).

Eukaryote Shannon indices were significantly different between sediments and water 0.2 µm (W2 > S) (Appendix A). The same was observed for ASV richness (W2 > S). No significant differences were observed for evenness.

### 3.3. Taxonomic Composition of the Microbial Communities

The surface soils were composed of a majority of *Actinobacteriota* and *Proteobacteria* for the Bacteria domain; *Crenarchaeota*, *Thermoplasmatota*, and *Nanoarchaeota* for the Archaea domain; and *Obazoa*, TSAR and *Archaeplastida* for the Eukaryote domain (Figure 2a, Figure 3a and Figure 4a). The cave sediments were dominated by *Proteobacteria*, *Actinobacteriota*, *Gemmatimonadota* and *Acidobacteria* at the phylum level for the bacteria. At the genus level, the main bacterial taxa were *Rhodoferax*, *Nitrospira*, unclassified (unc.) TRA3-20, unc. *Burkhloderiales*, unc. wb1-P19 and unc. *Gemmatimonadaceae* (Figure 2b). The archaea were dominated by *Thermoplasmatota* and *Crenarchaeota* at the phylum level and unc. *Methanomassiliicoccales*, unc. *Nitrosopumilaceae*, unc. *Nitrosotaleaceae*, unc. *Nitrososphaeria* and SCGC AAA011-D5 *Nanoarchaeia* at the genus level (Figure 3b). The eukaryotes were dominated by Obazoa and TSAR at the phylum level, and *Glissomonadida*, unc. *Mortierellaceae*, *Pezizomycotina*, unc. *Fungi*, *Pansomonadida*, *Blastocladiomycotina*, *Chelicerata* and *Lobosa* at the sub-phylum level (Figure 4b).

The 0.2 µm cave water was composed of a majority of *Proteobacteria*, *Bacteroidota* and *Actinobacteriota* at the phylum level for the Bacteria domain. At the genus level, the main bacterial taxa were *Limnohabitans*, *Prevotella* 9, *Nitrospira*, *Methylotenera* and hgcI clade. The archaea were dominated by *Crenarchaeota* and *Nanoarchaeota* at the phyum level, and *Nitrosarchaeum*, unc. *Woesearchaeales*, GW2011_GWC1_47_15 *Nanoarchaeota* and unc. Marine Group II *Thermoplasmata* at the genus level. The eukaryotes were dominated by *Obazoa*, TSAR and *Cryptista* at the phylum level, and *Cyclopoida*, *Cryptomonadales*, *Hymenostomatia*, *Rozellomycota*, *Gastrotricha* and *Glissomonadida* at the sub-phylum level.

The 0.1 µm cave water was composed of a majority of *Patescibacteria*, *Proteobacteria* and *Bdellovibrionota* at the phylum level for the bacteria. At the genus level, the main bacterial taxa were unc. *Saccharimonadales*, 0319-6G20 *Oligoflexia*, *Silvanigrella*, *Shewanella* and *Bdellovibrio*. The archaea were dominated by the *Nanoarchaeota* at the phylum level, and unc. *Woesearchaeales*, GW2011_GWC1_47_15 *Nanoarchaeota* and SCGC AAA011-D5 *Nanoarchaeia* at the genus level.

### 3.4. Beta-Diversity

Ordinations of bacterial, archaeal and eukaryote communities showed clusters of samples based on habitat type (surface soils, surface water, cave sediments, cave water 0.2 and 0.1 µm) (Figure 5a–c). For the bacteria, the cave sediment samples clustered closer to the surface soils and water samples than the cave water samples. The cave 0.2 µm water samples were the most dissimilar compared to the other sample clusters. For the archaea, the cave 0.2 µm water samples were also the most dissimilar compared to the other sample clusters. For the eukaryotes, all three sample groups (surface sediments, cave sediment and 0.2 µm water) were all clustered distinctively.

PERMANOVA analyses effectively confirmed the compositional difference in the communities of Bacteria, Archaea, and Eukaryote domains between the three habitat types (Appendix A). Habitat type explained 51.3% of the community variance for the Bacteria, 63.3% for the Archaea and 39.4% for the Eukaryote domains. AMOVA analyses comparing each habitat community 1 to 1 confirmed that all communities were significantly different from each other for each domain (Appendix A). HOMOVA analyses comparing the cave 0.2 and 0.1 µm water communities showed that, for the Bacteria domain, the 0.1 µm community had a significantly larger amount of variation compared to the 0.2 µm community (Appendix A). The same was observed for the Archaea domain.

### 3.5. Beta-Diversity and Correlation with Environmental Parameters

The surface soils and sediments were analyzed separately from the water since the measured environmental variables (pH, Ctot, Corg, Cinorg, Ntot and δ^13^C) were different from those measured in the cave water samples (DIC, DOC, NHx and NO_3_^−^). The 0.2 µm water was analyzed separately from the 0.1 µm water since the measured values were the same for both communities belonging to the same sample.

The surface soils and cave sediments were significantly correlated with pH and total nitrogen for the Bacteria domain (Appendix A) explaining 2.1 and 6% of the community variance, and with pH for the Archaea and Eukaryote domains explaining 4.5 and 3.9% of the community variance. The 0.2 µm water communities were significantly correlated with DOC for the bacteria (explaining 3.4% of the community variance) and with DIC and NO_3_^−^ for the archaea (explaining 8.2 and 7.3% of the community variance). There was no significant correlation for the eukaryotes, nor for any domain for the 0.1 µm communities. For the bacteria, the db-RDA graph showed that the surface soils were correlated with a higher total nitrogen content, while a higher pH was associated with the cave sediments (Appendix A). The graphs for the Archaea and Eukaryote domains also showed that the cave sediments were correlated with a higher pH (Appendix A).

### 3.6. Discriminative Microbial Taxa between Sample Groups

For the bacteria, at the genus level, when comparing all four sample groups (surface soils, cave sediments and cave water 0.2 and 0.1 µm) together, we observed using a LEfSe analysis that *Kribbella*, *Nocardioides*, *Mycobacterium*, 67-14 *Solirubrobacterales* and unc. *Xanthobacteraceae* were among the genera that were significantly more prevalent in the surface soils (Figure 6a). TRA3-20 *Burkholderiales*, unc. *Burkholderiales*, unc. *Gemmatimonadaceae*, IS-44 *Nitrosomonadaceae* and Subgroup 2 *Acidobacteriae* were significantly associated with cave sediments. *Limnohabitans*, *Prevotella* 9, *Methylotenera*, hgcI clade *Sporichthyaceae* and *Bacteroides* were significantly associated with 0.2 µm cave water. Unc. *Saccharimonadales*, 0319-6G20 *Oligoflexia*, *Silvanigrella*, *Bdellovibrio* and LWQ8 *Saccharimonadales* were significantly associated with 0.1 µm cave water.

For the archaea, unc. *Nitrososphaeraceae* and cand. *Nitrosocosmicus* were significantly more prevalent in the surface soils (Figure 6b). Unc. *Methanomassiliicoccales*, unc. *Nitrosopumilaceae*, unc. *Nitrosotaleaceae*, unc. *Nitrososphaeria* and Group 1.1c *Nitrososphaeria* were significantly associated with cave sediments. *Nitrosarchaeum*, cand. *Nitrosotenuis*, *Methanoregula* and CG1-02-32-21 *Micrarchaeales* were significantly associated with 0.2 µm cave water. GW2011_GWC1_47_15 *Nanoarchaeota*, unc. *Woesearchaeales*, SCGC AAA011-D5 *Nanoarchaeota* and cand. *Iainarchaeum* were significantly associated with 0.1 µm cave water.

For the eukaryotes, *Pezizomycotina*, *Hypotrichia*, *Annelida*, *Chromadorea* and *Colpodea* were among the significantly more prevalent sub-phylum taxa in the surface soils (Figure 6c). *Chelicerata* was significantly associated with cave sediments. *Hymenostomatia*, *Rozellomycota*, unc. *Alveolata*, *Eimeriida* and *Ochromonadales* were significantly associated with 0.2 µm cave water.

### 3.7. Microbial Source Tracking

For the cave sediment communities, we used as potential source communities all surface soil samples, the surface water, the other sediment samples, and all cave water samples (0.2 and 0.1 µm). For the cave water 0.2 µm communities, we used as potential sources communities from all surface soil samples, the surface water, all cave sediment samples and all cave water samples apart from the analyzed sample. We did the same for the 0.1 µm communities. To simplify data visualization and analyses, we merged as single sources all surface soil samples, all sediment samples, all 0.2 µm water samples and all 0.1 µm water samples. 

For the bacteria in cave sediments, most of the communities originated from the other sediment communities (between 25 and 35%) (Figure 7a). For the S1 sample (Radiesthesia gallery), 1.9% also stemmed from the surface soils. Apart from the S4 sample, the water samples contributed less than 1% of the sediment communities. The source of the cave water 0.2 µm communities was mainly other 0.2 µm communities (more than 75%) with some contribution from the 0.1 µm communities (between 1.4 and 4%) (Figure 7b). The W3 sample had a low contribution from the surface water (0.96%). The source of the cave water 0.1 µm communities was mainly other 0.1 µm communities (between 23 and 73%) (Figure 7c). Apart from sample W6, the sediment communities also contributed (between 0.36 and 8.4%), as well as the 0.2 µm communities (between 2.1 and 4.5%).

For the archaea in cave sediments, most of the communities originated from the other sediment communities (between 11 and 74%) (Figure 8a). The water samples contributed less than 1% of the sediment communities. The source of the cave water 0.2 µm communities was mainly other 0.2 µm communities (more than 83%), with some contribution from the 0.1 µm communities (between 2.5 and 9.5%) (Figure 8b). The W7 sample had a contribution from the surface soils (7.49%). The source of the cave water 0.1 µm communities was mainly other 0.1 µm communities (between 38 and 81%) (Figure 8c), but the sediment communities also contributed (less than 1%), as well as the 0.2 µm communities (between 2.4 and 9.7%).

For the eukaryotes in cave sediments, part of the communities originated from the other sediment communities (between 9.3 and 41.1%) (Figure 9a). Between 2.4 and 12.1% of the sediment communities stemmed from the surface soil communities. The source of the cave water 0.2 µm communities was mainly other 0.2 µm communities (between 31.7 and 87.6%) (Figure 9b).

## 4. Discussion

### 4.1. Environmental Properties and Geochemical Connectivity of the Saint-Leonard Cave

Like in most documented caves, the Saint-Leonard groundwater can be classified as oligotrophic based on its measured mean DOC value of 1.6 mg/L [85]. This value is just under the Canadian mean DOC value for wells (1.8 mg/L) [86] and below the global mean DOC value for groundwater (3.8 mg/L) [87]. However, DIC concentration (46 mg/L) was slightly higher than the global average for groundwater (30–43 mg/L) [88]. Furthermore, based on the mean value of total inorganic nitrogen content in the water (2.44 mg/L), Saint-Leonard cave’s groundwater could be considered eutrophic [89] or hypereutrophic [90]. The high concentration of nitrate could be explained by the urban setting of Saint-Leonard cave, combined with its low depth. This phenomenon has indeed been observed in three urban caves in the cities of Częstochowa and Kraków in Poland [91].

However, trophic classifications based on nitrogen availability for photosynthetic primary producers in lake ecosystems are not suitable for underground aquatic environments, devoid of any light source. The high concentrations of DIC could potentially sustain chemosynthesis-based autotrophy. Although chemoautotroph-driven communities have been found in a few caves [92,93,94,95], subterranean microbial communities are often dominated by heterotrophs that rely on the supply of allochthonous sources of carbon imported by percolating water [31,36,38,85,96].

The DOC found in the Saint-Leonard cave’s groundwater most probably traces its origins to the surface soils. Indeed, the δ^13^C measures for both the surface soils and the cave sediments displayed similar values, suggesting that the carbon found in the cave sediments originates from the surface soils. The δ^13^C range (between −27.9 and −24.4‰) matches that of C3 plants [97], which are found in temperate soil regions such as the Quebec province in Canada. The δ^15^N measures in surface soils (0.5 to 4‰) showed typical values associated with C3 plants [98]. The higher δ^15^N values for cave sediments (3.8 to 7‰) could potentially be attributed to the leaching of fertilizers or to a wastewater discharge [99], a hypothesis that would be supported by the high nitrate concentration in the groundwater. Overall, these results demonstrate that the above and belowground are geochemically connected.

### 4.2. Potential Biological Links between Surface Soils and the Cave Sediments and Water

Microbial source tracking indicated that the surface soil eukaryote communities contributed in a small capacity to the cave sediment communities. Apart from one sediment bacterial community and one 0.2 µm water archaeal community that showed a light contribution from the surface soils, bacterial and archaeal cave communities stemmed very little from the surface communities. Therefore, although there is strong evidence of a direct link between surface and cave water attested by hydrogeological data [60] as well as this study’s isotopic data showing a clear link between surface and cave organic matter, the environmental differences between both habitats are likely too strong to allow most surface communities seeping into the cave to survive [100].

We observed that bacterial community evenness was higher in surface soils than in cave sediments and water 0.1 µm, while in eukaryote communities, richness was higher in water 0.2 µm than in surface soils. Our results show that, despite their distinct environmental conditions, the surface soils and the cave’s interior habitat generally differ only slightly in terms of bacterial and eukaryote alpha-diversity indices. As for the Archaea domain, both cave water size fractions had Shannon and ASV richness indices significantly higher than in the surface soils. This suggests that archaea thrive more in, or are better adapted to, the conditions found in the cave’s groundwater than those found in the surface soils.

Our beta-diversity analyses revealed that habitat type explained more than half of the community compositional variance for the Bacteria and Archaea domains and 1/3 of the variance for the Eukaryote domain. Surface soils, cave sediments and groundwater all harbored distinct assemblages, suggesting an important environmental filtering process in the assembly of microbial communities likely related to the differences in abiotic conditions. The results from the db-RDA analyses suggested pH differences between both surface soils and cave sediments were the main driver for their distinct community compositions. Despite their distinctiveness, the taxonomic composition of these two habitats was more similar to each other than to those of groundwater for the Bacteria and Archaea domains, which was to be expected given their overall similar physical characteristics.

The bacterial genera that were more prevalent in the surface soils were typical soil and rhizosphere heterotrophs, notably *Kribbella*, *Nocardioides* and *Mycobacterium*, which all belong to the *Actinobacteriota* phylum predominant in soil [101]. Most species of those genera are mesophilic with an optimum growth around 30 °C and have a pH range from 5 to 9, often with an optimum at 7 [102,103,104,105,106,107,108].

The *Crenarchaeota* was by far the most dominant phylum in the surface soils for the archaea. Their prevalence was expected as those archaea are typically abundant in soils [109,110], although they are also commonly found in caves [42,43] as well as in aquifers [111]. Phylogeny of this archaeal group is in constant evolution, and mesophilic taxa have been reclassified in another phylum, *Thaumarchaeota* [112], and contain many obligate chemolithotrophs taxa that can oxidize ammonia in aerobic conditions [113,114,115]. Two *Crenarchaeota (Thaumarchaeota*) taxa of the *Nitrososphaeraceae* family, cand. *Nitrosocosmicus* and unc. *Nitrososphaeraceae*, were associated with surface soil communities. *Nitrososphaeraceae* is composed of aerobic chemolithoautotrophic archaea that can oxidize ammonia and fix CO_2_ [116], and are abundant in soils of karst ecosystems [117,118]. *Nitrosocosmicus* is also an ammonia-oxidizing archaeal genus [119,120] containing strains that have been isolated in near-neutral pH soils [121,122] as well as in municipal wastewater treatment plants [120].

Like in many cave habitats, eukaryote communities in surface soils were largely composed of *Obazoa* and TSAR clades, both highly diversified groups. The *Obazoa* clade encompasses the *Opishtokonta* clade mainly composed of *Metazoa* and *Fungi* [123]. *Metazoa* includes a plethora of eukaryote microbes like annelids, copepods, gastrotrichs, nematodes and rotifers [124] that can be found in soils as well as in caves. TSAR is a supergroup that includes *Stramenopiles*, *Alveolates* and *Rhizaria* (SAR) [125], as well as their sister clade *Telonemia* [126], composed of protists with extremely diverse morphologies, metabolisms and ecologies—including photosynthetic organisms, mixotrophs, heterotrophs, parasites and bacterivores [127]. *Pezizomycotina*, a subdivision of the *Ascomycota* phylum (fungi), was one of the most distinct eukaryote taxa in surface communities of the Pie-XII Park soils. *Ascomycota* is globally the most dominant phylum of fungal soil communities [128], and *Pezizomycotina* fungi are highly diverse in neutral pH temperate soils [129] like those of the Pie-XII Park. Those fungi can be bacterivores, saprophytes, endophytes or parasites and can form mycorrhizal associations as well as mutualistic associations with bacteria [130]. While *Pezizomycotina* is a typical fungus inhabiting caves [45], it was more strongly associated with soils in our study. *Hypotrichia* and *Colpodea*, both protist taxa from the *Ciliophora* phylum (TSAR), were also predominant in surface soils compared to the cave’s habitat. Hypotrichs and colpodeans are an important part of global soil biodiversity [131,132]. Unsurprisingly, the taxa *Annelida* (phylum) and *Chromadorea* (class of *Nematoda* phylum) were also prevalent in soils; segmented and roundworms play an important ecological role in urban parks and natural areas [133].

Overall, the slightly more alkaline pH in the cave sediments as well as the differences in nutrients and energy resources (absence of sunlight) could explain why the dominant soil genera were not as abundant inside the cave.

### 4.3. Sediment and Water Microbial Communities inside the Saint-Leonard Cave

The only difference in bacterial alpha diversity indices between the cave sediments and both water size communities was that the sediments harbored a higher number of ASVs compared to the 0.2 µm water. Furthermore, richness was higher in 0.1 µm water than in 0.2 µm water, but evenness was higher in 0.2 µm than in the 0.1 µm fraction. Overall, the pattern we observed in the distribution of bacterial taxa within communities of the cave’s interior habitats is that relatively rich communities have a low evenness, and vice versa. Saint-Leonard’s groundwater effectively contained few abundant bacterial taxa and many rare taxa—especially within the 0.1. µm fraction—a pattern frequently observed in groundwater microbiomes [134,135]. These rare taxa often play an essential role in ecosystem functioning, in biogeochemical cycles and, more generally, in functional diversity [136].

Archaeal communities showed a higher Shannon index in both water size fractions than in sediments, higher richness in 0.2 µm water than in sediments and higher evenness in 0.1 µm water than in both 0.2 µm water and sediments. These results showed that archaeal communities were generally more diverse in groundwater than in the sediments of the Saint-Leonard cave. Moreover, the high evenness of the ultra-small groundwater communities indicated a more evenly distributed taxonomic composition that could be due to smaller differences in competitive ability [137] and bigger importance of interspecific than intraspecific interactions in community function [138].

In terms of beta-diversity, our analyses showed that the different habitats (including the two size fractions of water) inside the Saint-Leonard cave harbored very taxonomically distinct bacterial, archaeal and eukaryote communities. Specifically, prokaryote communities living in the same type of ecological compartment were much more similar to one another than to communities from other compartments. Furthermore, we observed that ultra-small prokaryotic communities showed a larger amount of compositional variation than the communities represented in the 0.2 µm water fraction, which supported more homogeneous communities with similar alpha diversity and taxonomic compositions. In a study on eight karstic caves, Zhu et al. [139] also observed the influence of habitat type on compositional turnover. The correlation between shared habitat and community similarity could be explained by selection, a deterministic process that drives the assemblage of communities by favoring taxa that are better adapted to local abiotic and biotic conditions [140,141]. However, ecological stochastic processes can sometimes play a more important role in assembling prokaryote and eukaryote communities [100]. A number of studies have shown the importance of habitat and diverse environmental factors as selecting forces acting on microbial communities [142,143,144], notably in aquifers [145]. In belowground aquatic environments, hydrogeology is one of the key factors determining which ecological process dominates community assembly [134]. Compositional homogeneity and stability of the 0.2 µm size fraction of aquatic prokaryote communities could potentially be explained by the homogeneity in physicochemical conditions of the cave’s groundwater—hinting at the process of homogeneous selection driven by an environment with spatially uniform conditions [146,147]. Homogeneous selection is in fact a dominant process in certain aquatic [148] and sedimentary [149] ecosystems.

Our microbial source tracking analyses suggested some exchange between communities from the different cave compartments. Contributions from the 0.2 µm size fraction groundwater communities to the 0.1 µm size fraction communities and vice versa ranged from 1 to 10% for the Bacteria and Archaea domains. Nonetheless, a major part of the community for each domain and compartment seems to have originated from the other areas of the same compartment, supporting our assumption that local abiotic conditions—selection—likely shape the assembly and evolution of the microbial communities.

Unsurprisingly, *Proteobacteria* was the most dominant bacterial phylum in the Saint-Leonard groundwater and sediments—apart from the ultra-small bacterial phyla being relatively more abundant in the 0.1 µm size fraction of water. *Proteobacteria* is a ubiquitous phylum and one of the most abundant in caves where it is found in every ecological compartment [41,139]. Members of this phylum possess a large metabolic diversity and can catabolize a vast array of organic compounds [42]. *Acidobacteria* and *Gemmatimonadota*, two other phyla often found dominating cave sediments [41,150,151], also accounted for a good proportion of the bacteria and contributed to habitat-distinctive taxa. Differences in bacterial community composition for the cave sediments were explained by a higher proportion of TRA3-20 *Burkholderiales,* unc. *Burkholderiales* and IS-44 *Nitrosomonadaceae* (*Proteobacteria*), as well as unc. Subgroup 2 *Acidobacteriae* (*Acidobacteriota*) and unc. *Gemmatimonadaceae* (*Gemmatimonadota*). TRA3-20 is an uncultured bacterium associated with agricultural soils [152] and is also found in lake sediments [153] and plant litter from the city of Montreal [154]. This bacterium was identified as a potential keystone taxon involved in carbon mineralization and is likely to also be a major player in carbon cycling in the Saint-Leonard sediments. Unc. *Methanomassiliicoccales* (archaea), consisting of strictly anaerobic dihydrogen-dependent methanogens [155], were also strong drivers of the compositional difference in the cave sediments. They are part of the *Thermoplasmatota*, ubiquitous archaea typically abundant in cave compartments, especially sediments [156], and probably involved in carbon mineralization in the sediments as well. Many bacteria and archaea genera associated with the cave sediments belong to ammonia-oxidizing groups (*Nitrosomonadaceae,* unc. *Nitrosopumilaceae*, unc. *Nitrosotaleaceae* and unc. *Nitrososphaeria*), highlighting nitrogen-cycling activities within the cave, possibly due to the links with surface urban soils. For eukaryote communities inside the Saint-Leonard cave, the *Chelicerata* (*Obazoa*) subphylum, a clade of arthropods, explained the difference in the sediment communities. It might possibly be attributable to the presence of troglomorphic spiders, an important component of hypogean animal faunal communities [157]. Acari and Collembola have been shown to be microbivores attracted to microbial cells in Slovakian caves [158]. The same food web pattern can probably occur in the Saint-Leonard cave sediments.

Differences in community composition for the 0.2 µm size fraction groundwater communities were explained by two *Proteobacteria* genera, *Limnohabitans* and *Methylotenera*; two genera from the *Bacteroidota* phylum, *Prevotella* 9 and *Bacteroides*; and hgcI clade *Sporichthyaceae* (*Actinobacteriota* phylum). *Bacteroidota*, which made up substantial proportion of the cave’s water bacterial communities, is a ubiquitous phylum [159] and has been found to dominate groundwater assemblages [160]. *Actinobacteriota* also accounted for a relatively large portion of the cave’s water and sediment communities, corroborating previous studies [40,41,42]. Interestingly, members of this phylum residing in caves are considered a promising source of novel antibiotics for humans [161,162]. *Prevotella* 9 and *Bacteroides* are both associated with the human gut and feces and are probably a sign of wastewater or sewage seeping into the cave waters [163,164]. *Limnohabitans* and hgcI are common freshwater heterotrophic bacteria [165,166,167]. Furthermore, the hgcI *Sporichthyaceae* are predicted to have the ability to use inorganic nutrients and nitrogen-rich organic compounds [168,169], all of which were high in the Saint-Leonard cave groundwater and significantly correlated with bacterial community composition. *Methylotenera* is a methylotroph able to use methane-derived carbon in eutrophic lakes [170]. The presence of these C_1_-utilizing bacteria is likely linked to the detection of methane-producer *Methanoregula.* The CG1-02-32-21 *Micrarchaeales* from the phylum *Micrarchaeota* (DPANN superphylum), which is found in all types of environments, including groundwater [171], was part of the drivers of the 0.2 µm size fraction archaeal groundwater community and is a potential complex organic carbon utilizer [172]. For eukaryote communities, another *Obazoa* taxa, *Rozellomycota*, a basal or sister clade of fungi, drove the distinctiveness of 0.2 µm water communities. These organisms are parasites of amoebae and also algae and small invertebrates [173]. Along with *Rozellomycota*, many TSAR taxa were also associated with the 0.2 µm water, among which are *Hymenostomatia*, unc. *Alveolata*, *Eimeriida* and *Ochromonadales.* Being highly diverse, the TSAR supergroup is typically strongly present in microbial eukaryote communities in cave ecosystems [46]. It has been proposed that sediments might serve as a refuge habitat for cyst-forming protists and act as “seed banks” to recolonize groundwater [174].

In the 0.1 µm size fraction groundwater, two ultra-small bacterial phyla dominated the communities: *Patescibacteria* and *Bdellovibrionota*. Differences in community composition were explained by a higher proportion of unc. *Saccharimonadales* and LWQ8 *Saccharimonadales* (*Patescibacteria*), as well as 0319-6G20 *Oligoflexia*, *Silvanigrella*, *Bdellovibri* (*Bdellovibrionota*). *Patescibacteria*, which includes a large part of the candidate phyla radiation (CPR), is a superphylum of ultra-small bacteria found in high numbers in groundwater habitats [134,175]. Members of *Patescibacteria* adapted to this environment have a streamlined genome that shows a reduction of many non-essential metabolic functions, which suggests the necessity of engaging in symbiotic or syntrophic interactions to acquire nutrients—notably via pili [16,176]. In *Patescibacteria*, horizontal gene transfer seems to be an important mechanism of genome adaptation in subsurface aquatic environments [177]. *Bdellovibrionota* bacteria are pleiomorphs and some have an ultra-small size [178]. Groundwater is a choice habitat for *Bdellovibrionota* [179]. These predatory bacteria are obligate bacterivores [180,181]. Therefore, these microorganisms have a direct impact on the structure of bacterial communities—they add an extra layer of complexity to the microbial loop and to the recycling process of organic matter and nutrients [182]. *Nanoarchaeota*, another characteristic archaeal phylum of groundwater microbial communities [183], formed an important part of the caves’ groundwater microbiome, especially in the 0.1 µm size fraction where it largely dominated the communities and harbored distinctive taxa such as GW2011_GWC1_47_15 *Nanoarchaeota*, unc. *Woesearchaeales* and SCGC AAA011-D5 *Nanoarchaeia*. *Nanoarchaeota* is also part of the DPANN radiation and possesses all typical characteristics of ultra-small prokaryotes [184,185]. Apart from groundwater, these archaea are also found in extreme environments like hydrothermal vent sediments, hypersaline sediments [186] and acidic hot springs [187]. Archaea of this phylum are obligate ectoparasites of other archaea [188]. These findings suggest the existence of complex symbiotic and trophic interactions, which most probably impact the community structure of aquatic communities in the Saint-Leonard cave.

## 5. Conclusions

This study is the first to investigate microbial communities residing inside the Saint-Leonard cave—one of the extremely rare urban glaciotectonic caves in the world. Inhabiting its groundwater and sediments, we detected a microbial biodiversity typical of cave ecosystems, with habitat-distinctive bacterial, archaeal and eukaryote communities. Taxa that we detected were associated with a wide variety of lifestyles and metabolic capacities. While we found evidence that the organic matter and nutrients originated from the surface soil above the cave, our results suggest that habitat selection, rather than dispersal between habitat types, was probably driving the community assembly. Our study has revealed that the cave’s groundwater is generally richer in microbial taxa than sediments. Moreover, it harbors a considerable proportion of ultra-small bacteria and archaea from diverse prokaryote phyla such as *Bdellovibrionota*, *Patescibacteria*, *Woesearchaeota* and *Nanoarchaeota*. The observation that 0.1 µm-filtered (and other) samples harbored a noticeable fraction of unclassified bacteria and archaea reflects the fact that these groundwater communities, notably the ultra-small prokaryotes, need to be better studied, characterized and considered when assessing the taxonomic microbiomes of natural environments. Our results open the door to a functional study of the cave’s microbiomes to unveil key metabolisms that would link together the taxonomic structure of communities, the microbial functions and the physicochemical characteristics of the cave’s habitats.

## Figures and Tables

**Figure 1 microorganisms-12-01791-f001:**
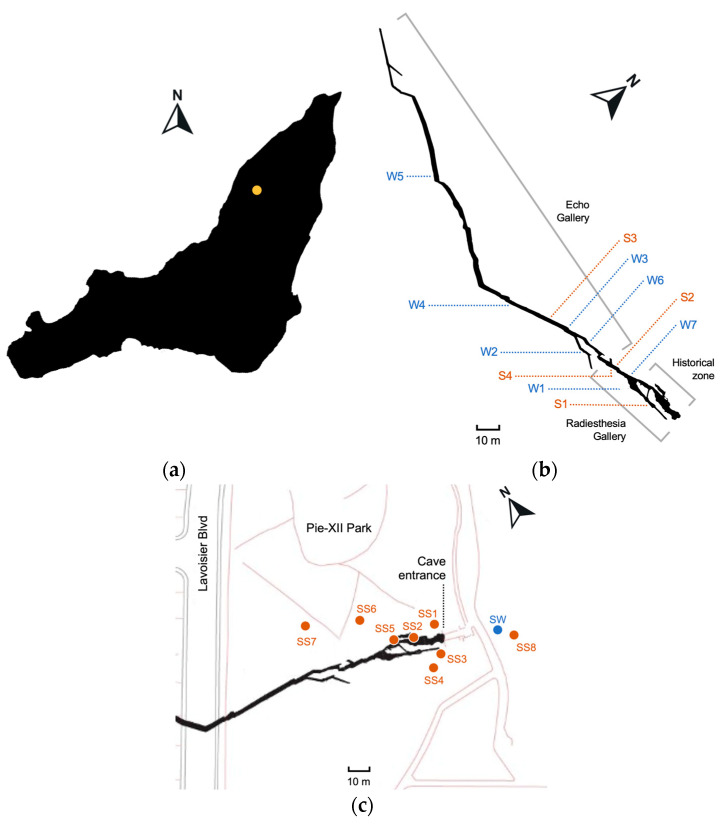
Geographic position of the Saint-Leonard cave (yellow dot) on the island of Montreal, Canada (**a**). Location of the water (W) and sediment (S) samples inside the Saint-Leonard cave (**b**) and of the surface soil (SS) and surface water (SW) samples outside the cave in the Pie-XII Park (**c**).

**Figure 2 microorganisms-12-01791-f002:**
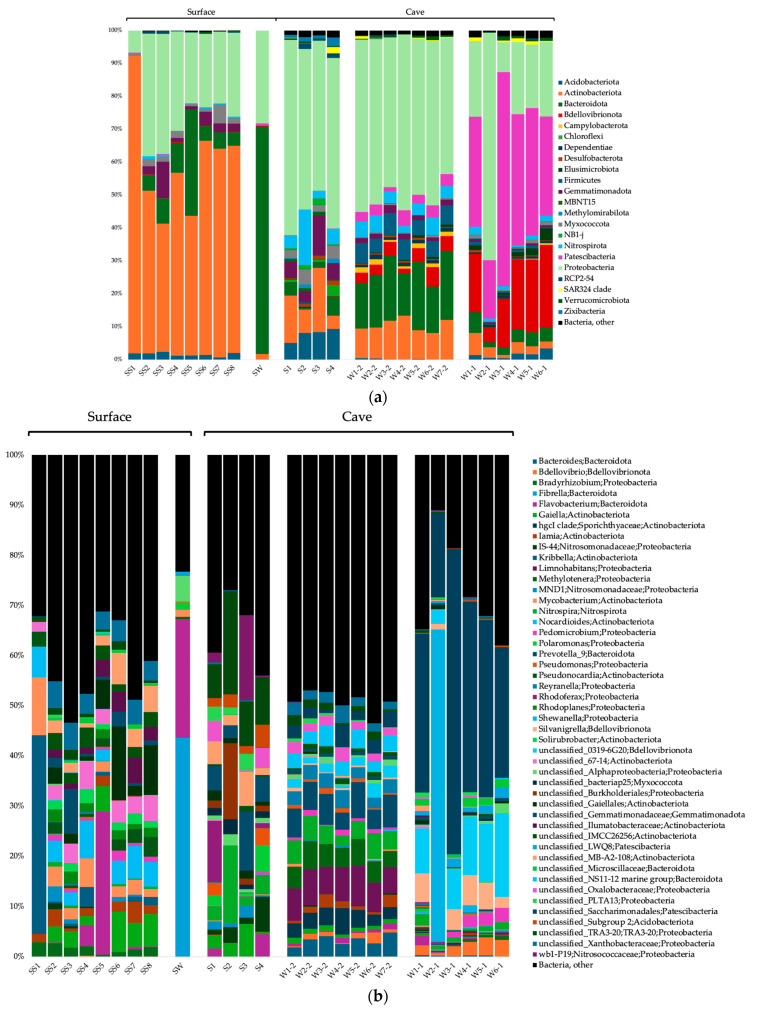
Taxonomical identification of the 16S rRNA gene sequences for the Bacteria domain at the phylum level (**a**) and genus level (**b**). Only phyla with ≥1% relative abundance are shown. SS, surface soils; S, sediments; W#-1, water 0.1 µm; W#-2, water 0.2 µm; SW, surface water.

**Figure 3 microorganisms-12-01791-f003:**
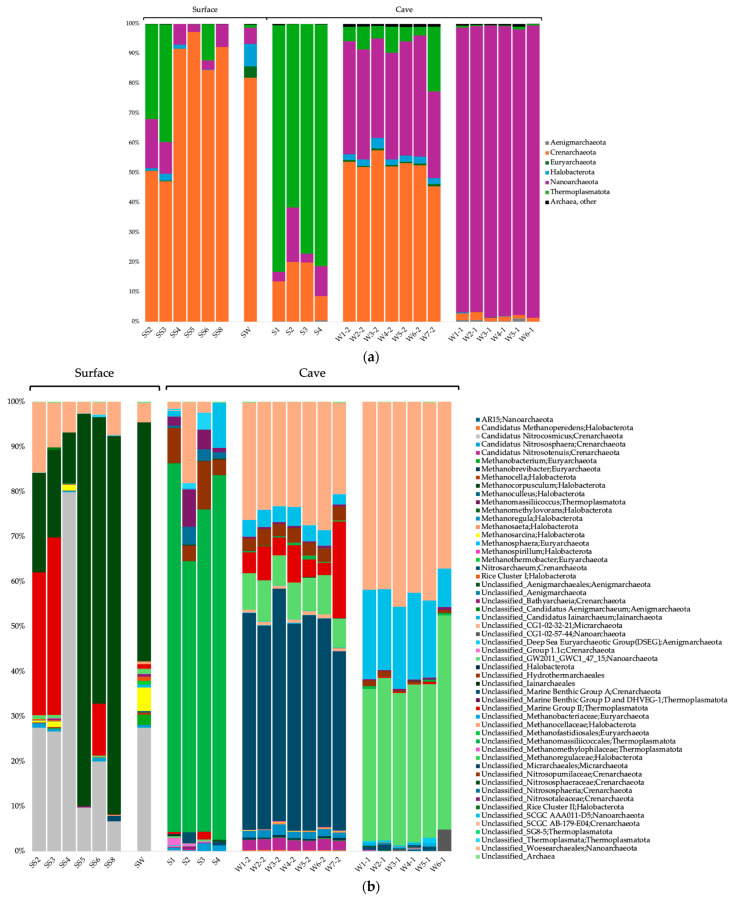
Taxonomical identification of the 16S rRNA gene sequences for the Archaea domain at the phylum level (**a**) and genus level (**b**). Only phyla with ≥1% relative abundance are shown. SS, surface soils; S, sediments; W#-1, water 0.1 µm; W#-2, water 0.2 µm; SW, surface water.

**Figure 4 microorganisms-12-01791-f004:**
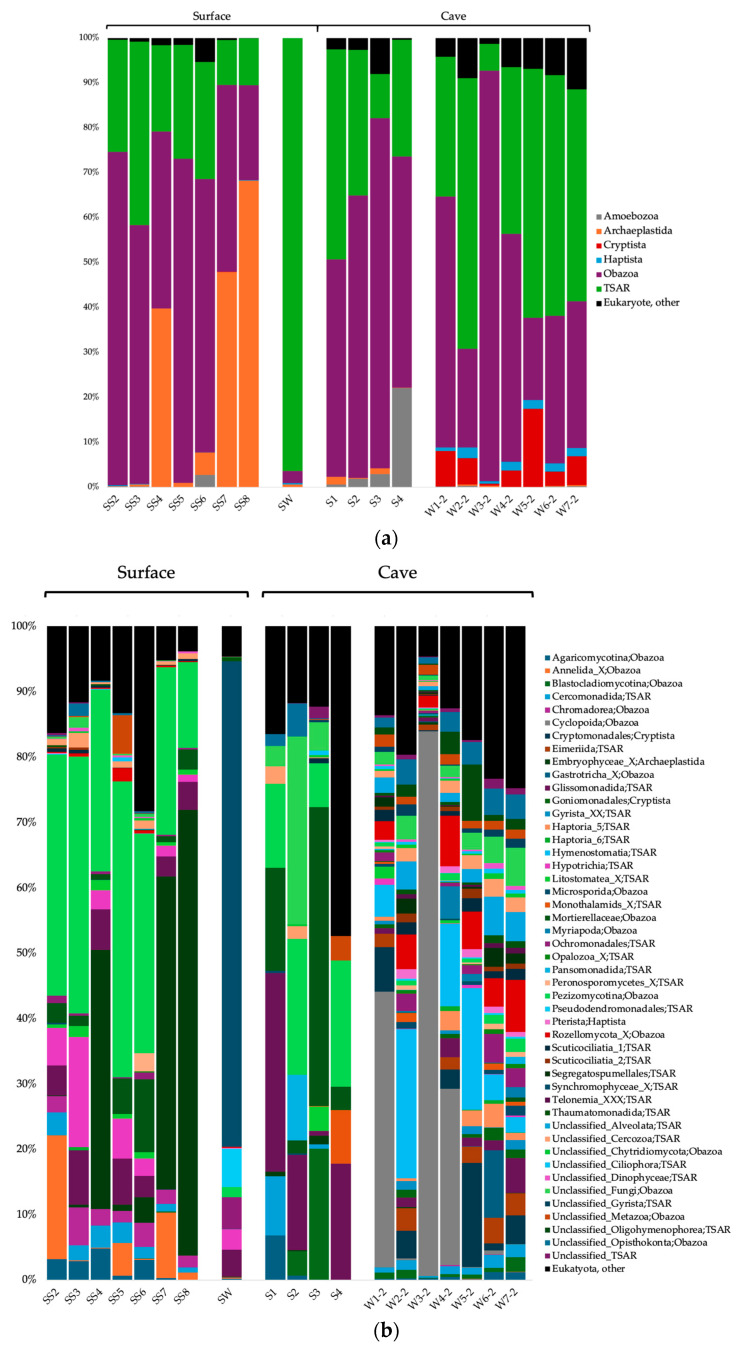
Taxonomical identification of the 18S rRNA gene sequences for the Eukaryote domain at the phylum level (**a**) and sub-phylum level (**b**). Only phyla with ≥1% relative abundance are shown. SS, surface soils; S, sediments; W#-2, water 0.2 µm; SW, surface water.

**Figure 5 microorganisms-12-01791-f005:**
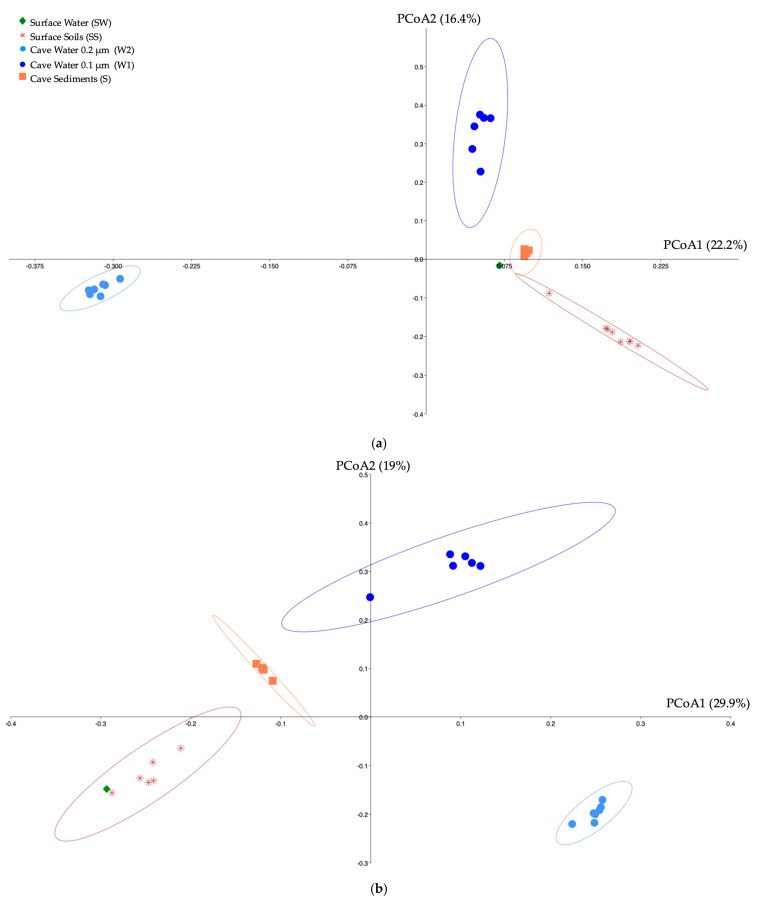
Principal Coordinate Analysis (PCoA) ordination of bacterial (**a**), archaeal (**b**), and eukaryote (**c**) community composition based on a Bray–Curtis dissimilarity matrix. Percentages indicate proportion of variance explained for the first two axes. Ellipses show 95% confidence intervals by habitat.

**Figure 6 microorganisms-12-01791-f006:**
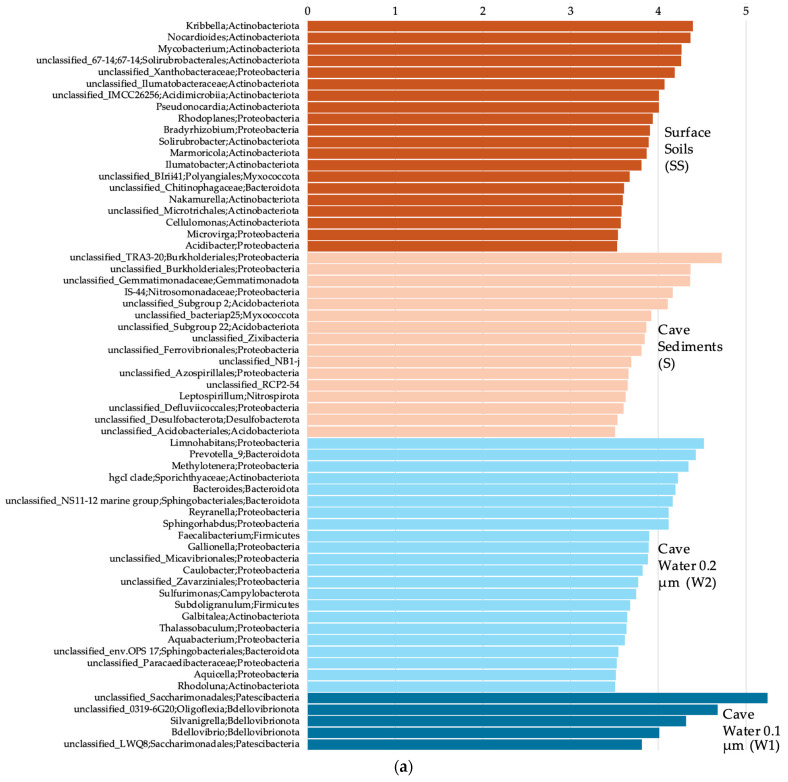
Linear discriminant analysis (LDA) score showing genera or sub-phylum taxa significantly associated with each habitat (log LDA score ≥ 3.5, *p* < 0.05), for the Bacteria (**a**), Archaea (**b**), and Eukaryote (**c**) domains.

**Figure 7 microorganisms-12-01791-f007:**
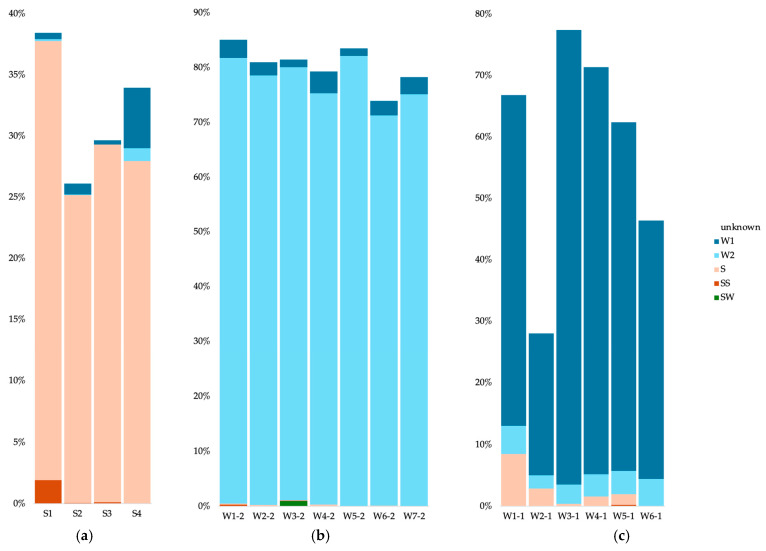
Microbial source tracking for the bacteria, using FEAST, for the cave sediment communities (**a**), cave 0.2 µm water communities (**b**) and cave 0.1 µm water communities (**c**). SW, surface water; SS, surface soils; S, sediments; W#-2, water 0.2 µm; W#-1, water 0.1 µm.

**Figure 8 microorganisms-12-01791-f008:**
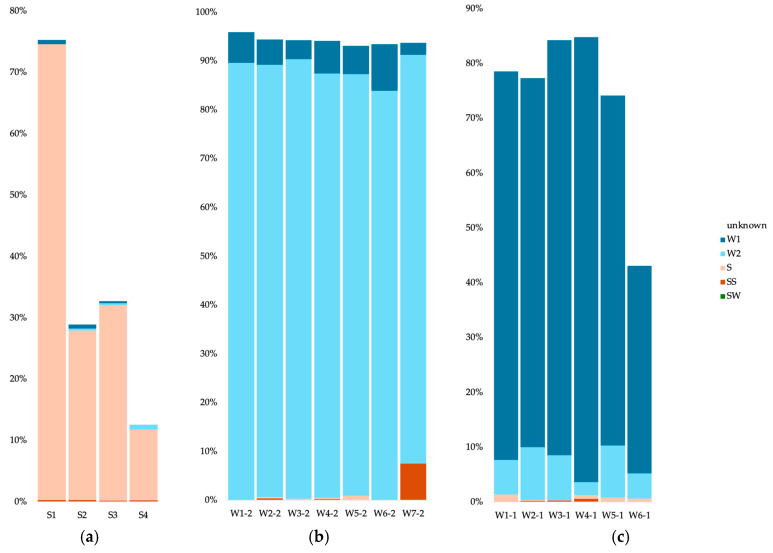
Microbial source tracking for the archaea, using FEAST, for the cave sediment communities (**a**), cave 0.2 µm water communities (**b**) and cave 0.1 µm water communities (**c**). SW, surface water; SS, surface soils; S, sediments; W#-2, water 0.2 µm; W#-1, water 0.1 µm.

**Figure 9 microorganisms-12-01791-f009:**
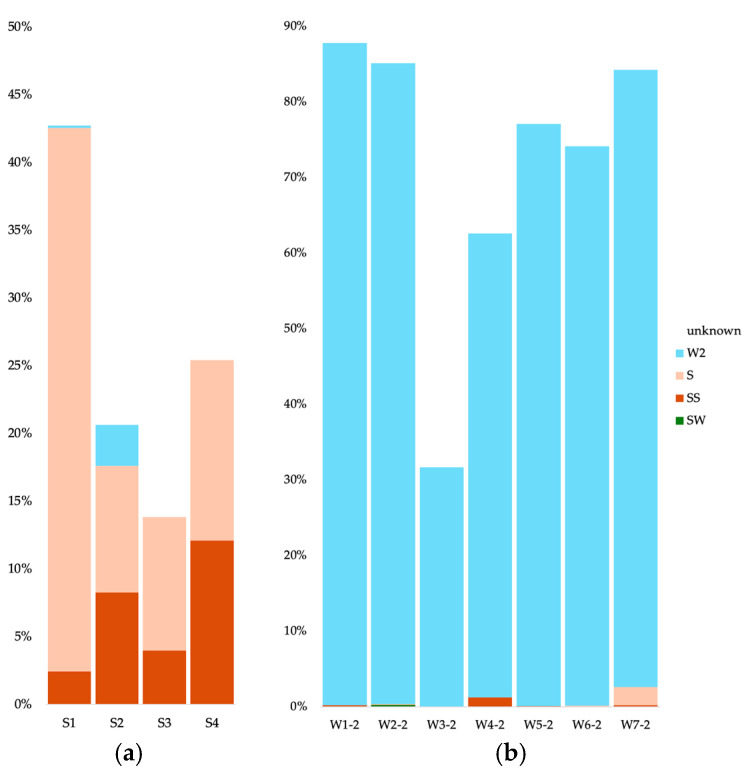
Microbial source tracking for the eukaryotes, using FEAST, for the cave sediment communities (**a**) and cave 0.2 µm water communities (**b**). SW, surface water; SS, surface soils; S, sediments; W#-2, water 0.2 µm.

**Table 1 microorganisms-12-01791-t001:** Groundwater geo- and physicochemical properties. All parameters apart from pH are expressed in mg/L. DOC, dissolved organic carbon; DIC, dissolved inorganic carbon; NHx, ammonia/ammoniac; na, not applicable.

Water Samples	pH	DOC	DIC	NHx	NO_2_^−^	NO_3_^−^
Radiesthesia						
W1	7.8	1.7	46.81	0.0715	0.01	2.43
W7	7.4	1.6	43.89	0.0235	0.01	2.42
Echo						
W2	7.5	1.67	46.11	0.0325	0.01	2.43
W3	7.6	1.59	48.22	0.0275	0.01	2.44
W4	7.5	1.82	45.09	0.0275	0.01	2.42
W5	7.5	1.57	46.72	0.022	0.01	2.43
W6	7.5	1.53	45.13	0.027	0.01	2.45

**Table 2 microorganisms-12-01791-t002:** Cave sediment and external surface soil properties. All parameters apart from pH and isotopic values (‰) are expressed in mg/L. Rad, Radiesthesia; tot, total; org, organic; inorg, inorganic.

Samples	pH	Ctot	Corg	Cinorg	Ntot	δ^13^C	δ^15^N
Surface Soil							
SS1	5.8	28.05	27.88	0.17	0.95	−24.4	0.5
SS2	7.5	9.68	9.68	0	0.51	−28.1	2.0
SS3	7.1	11.18	11.18	0	0.49	−27.3	1.9
SS4	7.3	17.96	17.76	0.2	0.68	−25.9	0.6
SS5	7.5	10.64	10.09	0.55	0.88	−27.0	4.0
SS6	7.8	4.87	4.87	0	0.36	−27.9	3.9
Sediment							
S1 (Rad)	7.9	0.39	0.23	0.17	0.03	−24.8	na
S2 (Rad)	8.1	1.21	1.11	0.10	0.15	−24.9	7.0
S3 (Echo)	8.1	0.96	0.17	0.79	0.03	−24.7	na
S4 (Rad)	8.3	0.36	0.18	0.18	0.05	−26.3	3.8

## Data Availability

The original raw sequences analyzed in this study are openly available at the National Center for Biotechnology Information (NCBI) under the BioProject ID PRJNA1139756.

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
