# Peer review of "The Saint-Leonard Urban Glaciotectonic Cave Harbors Rich and Diverse Planktonic and Sedimentary Microbial Communities"

_microorganisms, 2024, doi:10.3390/microorganisms12091791_

Round 1
Reviewer 1 Report
Comments and Suggestions for Authors
The authors investigated the microbial community composition in an urban glaciotectonic cave by using the amplicon sequencing. The findings are intriguing and the writing is well-organized. I have a few suggestions for the MS to improve the quality.
1. The title looks a little bit ‘boring’, please modify it.
2. Abstract, lines 14-15, this statement is too strong. In fact, there have been reports on microbial composition in the caves. Thus, add the exact name of the cave.
3. Figure 1, please could you add one more picture clarifying where is the location of the Saint-Leonard cave? For example, how far away from the Montreal city?
4. Figure 2a, too many phyla presented. You can list the most fundamental ones. For example, relative abundance > 1% or else.
5. Figure 5. The quality is not good enough, not clear. The numbers are too tiny.
6. Table 3, I’m not sure this table is necessary here.
7. Section 3.6, the authors conducted LEfSe analysis, that’s good. I strongly suggest to further perform a KEGG enrichment analysis on these unique taxa in each habitat. We could know the metabolic functions of these microbes and speculate what pathways could be happened in each habitat. Besides, you also mentioned that “Communities were …. with a wide variety of lifestyles and of metabolic capacities” in the Abstract. The KEGG analysis is a good support for your statement.
8. In the discussion section, you can add more content about the metabolism of the microbial communities in the cave. The readers not only want to know what the microbes are, but are also curious about what they can do.
9. Conclusion is too clumsy and long. The second paragraph is not conclusion but the meaning of this paper. Move or delete it. And just list the key findings and the bullet statements of your work.
Author Response
The authors investigated the microbial community composition in an urban glaciotectonic cave by using the amplicon sequencing. The findings are intriguing and the writing is well-organized. I have a few suggestions for the MS to improve the quality.
Response from the authors: We would like to thank the reviewer for taking the time to read our manuscript and help improve it. You will find all revisions in track changes in the re-submitted word manuscript. In our replies below, we noted new line numbers from the word document (in track changes mode) and from the pdf version. Please make sure line numbers start at 1 in the word manuscript in track changes mode (if necessary, remove and add back the lines).
Comment 1: The title looks a little bit ‘boring’, please modify it.
Response from the authors: We changed the title.
Comment 2: Abstract, lines 14-15, this statement is too strong. In fact, there have been reports on microbial composition in the caves. Thus, add the exact name of the cave.
Response from the authors: This was added (line 14).
Comment 3: Figure 1, please could you add one more picture clarifying where is the location of the Saint-Leonard cave? For example, how far away from the Montreal city?
Response from the authors: We added a map showing the location of the Saint-Leonard cave on the island of Montreal (Figure 1a) (page 4). What used to be Figure 1a is now 1b, and what used to be Figure 1b is now 1c.
Comment 4: Figure 2a, too many phyla presented. You can list the most fundamental ones. For example, relative abundance > 1% or else.
Response from the authors: Figure 2a was modified (page 9). Only phyla with ≥1% relative abundance are now shown.
Comment 5: Figure 5. The quality is not good enough, not clear. The numbers are too tiny.
Response from the authors: Figure 5 was modified accordingly (pages 12-13).
Comment 6: Table 3, I’m not sure this table is necessary here.
Response from the authors: We moved this table to the supplementary material, Table S9 (Supp. mat. page 9).
Comment 7: Section 3.6, the authors conducted LEfSe analysis, that’s good. I strongly suggest to further perform a KEGG enrichment analysis on these unique taxa in each habitat. We could know the metabolic functions of these microbes and speculate what pathways could be happened in each habitat. Besides, you also mentioned that “Communities were …. with a wide variety of lifestyles and of metabolic capacities” in the Abstract. The KEGG analysis is a good support for your statement.
Response from the authors: We would argue here that genome prediction tools such as PICRUSt should not be used for this type of environmental sample, where the number of uncultured or unknown lineages is high. Furthermore, we actually went back to the cave and sampled more water for metagenomic and metatranscriptomic analyses. This data is currently being analyzed and will be presented in a separate paper. We focused here on the ecology aspect of the microbial communities in the cave, with their structure and possible links to surface communities.
Comment 8: In the discussion section, you can add more content about the metabolism of the microbial communities in the cave. The readers not only want to know what the microbes are, but are also curious about what they can do.
Response from the authors: As we wrote in our answer to the comment above, we wanted this paper to be more ecology-focused, which is why we used amplicon sequencing. However, we agree that it is very interesting for the reader to have a global picture of the functions of the cave’s microorganisms. That is the reason why we chose to present (in our first submitted manuscript) paragraphs in the discussion where we speculate on metabolisms based on the 16S/18S datasets (word lines L608-616, L621-630, L704-714, L727-740; pdf lines L580-588, L593-602, L676-686, L699-712). Also, because many taxa are uncultured or unknown at the genus, family or order level, it is sometimes difficult to discuss potential metabolisms.
Comment 9: Conclusion is too clumsy and long. The second paragraph is not conclusion but the meaning of this paper. Move or delete it. And just list the key findings and the bullet statements of your work.
Response from the authors: We deleted the second paragraph. We did very minor edits in the first paragraph. We believe that the first paragraph as it is written does state all of our key findings in an integrated manner, and highlights some important avenues for future studies.
Reviewer 2 Report
Comments and Suggestions for Authors
In this paper, Jocelyn Lauzon and colleagues present the results of profiling the microbial communities of bacteria, archaea and eukaryotes in the water and sediments of Saint-Leonard cave in Montreal city in Canada. In addition to the results of 16S/18S rRNA gene profiling, a data on the chemical composition of the samples was also obtained, which is an important advantage of the work. The main conclusion is that the microbial communities of the water and sediments of the cave are specific, despite the proximity of the cave to the surface and the influx of waters passing through the soil. All the data are well described and discussed in detail. I have one general question and few specific comments.
A general question concerns the consideration of possible contamination. As the authors write, “To decontaminate the ASV communities of each type of sample (water filtered at 0.2 and 0.1 μm, soils and sediments), we used the decontam package [77] with the kit blank control samples and the negative PCR controls”.
If the experiments are set up correctly and there is no laboratory contamination, the blank control and negative PCR control should not generate 16S/18S rRNA PCR product. However, in this work ”for each domain, a PCR negative control was sequenced” implying that PCR product was obtained. In this case, explanations are needed. How many ASV were detected for the control samples and what proportion of the rRNA gene reads in experimental samples (water, sediments) are represented by these ASV? If this share is small, then this is not a problem, but if contaminants make up a large part of the entire rRNA sequence dataset in the experimental samples, then all results became questionable.
Specific comments
lines 36-37. total darkness and anoxic conditions are certainly not extreme
lines 225-226. This is unclear. PCR products were not obtained?
Figures 2,3,4. Too many lineages are shown and it is difficult to identify them according to the color. I recommend to show only major lineages ( maximum 10) and combine the rest in “others” group.
Figure 5. The font sizes are too small. There is enough space to increase the font size so that it becomes like in the figure caption.
Author Response
In this paper, Jocelyn Lauzon and colleagues present the results of profiling the microbial communities of bacteria, archaea and eukaryotes in the water and sediments of Saint-Leonard cave in Montreal city in Canada. In addition to the results of 16S/18S rRNA gene profiling, a data on the chemical composition of the samples was also obtained, which is an important advantage of the work. The main conclusion is that the microbial communities of the water and sediments of the cave are specific, despite the proximity of the cave to the surface and the influx of waters passing through the soil. All the data are well described and discussed in detail. I have one general question and few specific comments.
Response from the authors: We would like to thank the reviewer for taking the time to read our manuscript and help improve it. You will find all revisions in track changes in the re-submitted word manuscript. In our replies below, we noted new line numbers from the word document (in track changes mode) and from the pdf version. Please make sure line numbers start at 1 in the word manuscript in track changes mode (if necessary, remove and add back the lines).
General question from reviewer: A general question concerns the consideration of possible contamination. As the authors write, “To decontaminate the ASV communities of each type of sample (water filtered at 0.2 and 0.1 μm, soils and sediments), we used the decontam package [77] with the kit blank control samples and the negative PCR controls”.
If the experiments are set up correctly and there is no laboratory contamination, the blank control and negative PCR control should not generate 16S/18S rRNA PCR product. However, in this work ”for each domain, a PCR negative control was sequenced” implying that PCR product was obtained. In this case, explanations are needed. How many ASV were detected for the control samples and what proportion of the rRNA gene reads in experimental samples (water, sediments) are represented by these ASV? If this share is small, then this is not a problem, but if contaminants make up a large part of the entire rRNA sequence dataset in the experimental samples, then all results became questionable.
Response from the authors: This is a relevant question to ask. Actually, we would argue here that no matter which precautions are taken, there is always some contamination in the PCR controls, especially when working with low microbial biomass samples like ours. This is why it has become routine to sequence the negative control of the PCR reactions used for sequencing. However, we do agree with the reviewer that if carried out correctly, they should not generate many sequences. In our case, for the Archaea, the negative PCR control had 0 reads, and the average of the total number of reads for the samples was 14,791 reads. For the Bacteria, the negative PCR control had 0 reads and the average of the total number of reads for the samples was 6,339 reads. For the Eukaryotes, the negative PCR control had 528 reads, and the average of the total number of reads for the samples was 9,111 reads. Using the decontam package, we detected some ASVs in the kit blank control samples (as well as the PCR control for the Eukaryotes), and we removed those ASVs for all analyses. In the manuscript (word lines 252-255; pdf lines 237-240), we added detailed information about the proportion of contamination as we believe it is a good suggestion from the reviewer to be transparent about it.
Specific comments from the reviewer:
lines 36-37. total darkness and anoxic conditions are certainly not extreme
Response from the authors: We changed the word ‘extreme’ to ‘harsh’ line 34.
lines 225-226. This is unclear. PCR products were not obtained?
Response from the authors: Thank you for pointing that out. An information did not get correctly relayed among us during the writing process. We effectively discarded the 0.1 µm water sample #7 for all analyses, not because of a low number of sequences, but because an error happened during the processing of this sample, upstream of the PCR, resulting in sequence composition extremely similar to 0.2 µm water samples, for the bacteria and archaea domains. Considering the very distinct community composition between 0.2 and 0.1 µm water fractions (see Figures 2, 3, and 5ab and PERMANOVA results), we are confident that sequences obtained for the water sample 0.1µm #7 can only be explained by a pre-PCR error and not by any biological phenomenon. We modified the manuscript to explain our choice of discarding this sample (word lines 243-245; pdf lines 228-230).
Figures 2,3,4. Too many lineages are shown and it is difficult to identify them according to the color. I recommend to show only major lineages ( maximum 10) and combine the rest in “others” group.
Response from the authors: We modified Figures 2a, 3a and 4a accordingly. Only phyla with ≥1% relative abundance are shown. We did not modify Figure 2b, 3b and 4b because different genera are abundant in different group samples and showing less lineages (e.g. 10 lineages) would not be representative.
Figure 5. The font sizes are too small. There is enough space to increase the font size so that it becomes like in the figure caption.
Response from the authors: Figure 5 was modified accordingly (pages 12-13).
Reviewer 3 Report
Comments and Suggestions for Authors
This is a substantial study documenting the complex interactions of biota between surface terrestrial conditions and those in the sub-terranean communities of a cave in Montreal Canada. Overall, the manuscript is well organized and clearly written. The biotic taxa analyzed are diverse including archaeal, bacterial, and a wide range of eukaryotes including microeukaryotes and members of the Obazoa, along with sufficient quantitative habitat variables to make a substantial contribution to our understanding of the environmental and biotic interactions that contribute to such a subterranean ecosystem, particularly highlighting the significant role of habitats. I found only a few recommended clarifications of some of the text.
Line recommended edits
85 “Owing to the myriads of metabolisms sustained by cave’s microorganisms---”
113 “Special attention was given ----” The initial A is not needed.
163 “All further analyses were conducted at ----”
221 A correction is needed in designating the truncated positions for the Eukaryotes (presently 260 and 260 is indicated). Do they mean perhaps 260 and 200 ? Please clarify.
314 “The Eukaryotes were dominated by the Obazoa and TSAR at the---"
323 “The Eukaryotes were dominated by the Obazoa, ------”
486 “For the Eukaryotes in cave sediments, part of ------”
551 “This suggests that Archaea thrive more in, or are better adapted to, the conditions found in the cave’s groundwater ---”
707 “Along with Rozellomycota, many TSAR taxa were also ----"
711 “---- as a refuge habitat for cyst-forming protists, and act ----”
742 “---- one of the extremely rare urban glaciotectonic caves in the world.”
Comments on the Quality of English Language
Only very minor corrections have been recommended to the authors.
Author Response
This is a substantial study documenting the complex interactions of biota between surface terrestrial conditions and those in the sub-terranean communities of a cave in Montreal Canada. Overall, the manuscript is well organized and clearly written. The biotic taxa analyzed are diverse including archaeal, bacterial, and a wide range of eukaryotes including microeukaryotes and members of the Obazoa, along with sufficient quantitative habitat variables to make a substantial contribution to our understanding of the environmental and biotic interactions that contribute to such a subterranean ecosystem, particularly highlighting the significant role of habitats. I found only a few recommended clarifications of some of the text.
Response from the authors: We would like to thank the reviewer for taking the time to read our manuscript and help improve it. You will find all revisions in track changes in the re-submitted word manuscript. In our replies below, we noted new line numbers from the word document (in track changes mode) and from the pdf version. Please make sure line numbers start at 1 in the word manuscript in track changes mode (if necessary, remove and add back the line numbers).
Recommended edits from reviewer:
L85: “Owing to the myriads of metabolisms sustained by cave’s microorganisms---”
Response from the authors: This was changed line 88 (pdf line 84).
L113: “Special attention was given ----” The initial A is not needed.
Response from the authors: This was changed line 117 (pdf line 112).
L163: “All further analyses were conducted at ----”
Response from the authors: This was changed line 175 (pdf line 167).
L221: A correction is needed in designating the truncated positions for the Eukaryotes (presently 260 and 260 is indicated). Do they mean perhaps 260 and 200 ? Please clarify.
Response from the authors: Indeed, we changed to “260 and 230” in the text, line 239 (pdf line 224).
L314: “The Eukaryotes were dominated by the Obazoa and TSAR at the---"
Response from the authors: This was changed line 342 (pdf line 321).
L323: “The Eukaryotes were dominated by the Obazoa, ------”
Response from the authors: This was changed line 352 (pdf line 331).
L486: “For the Eukaryotes in cave sediments, part of ------”
Response from the authors: This was changed line 522 (pdf line 494).
L551: “This suggests that Archaea thrive more in, or are better adapted to, the conditions found in the cave’s groundwater ---”
Response from the authors: This was changed line 587 (pdf line 559).
L707: “Along with Rozellomycota, many TSAR taxa were also ----"
Response from the authors: This was changed line 743 (pdf line 715).
L711: “---- as a refuge habitat for cyst-forming protists, and act ----”
Response from the authors: This was changed line 747 (pdf line 719).
L742: “---- one of the extremely rare urban glaciotectonic caves in the world.”
Response from the authors: This was changed line 779 (pdf line 750).
Round 2
Reviewer 1 Report
Comments and Suggestions for Authors
No comments